# Alpha1A- and Beta3-Adrenoceptors Interplay in Adipose Multipotent Mesenchymal Stromal Cells: A Novel Mechanism of Obesity-Driven Hypertension

**DOI:** 10.3390/cells12040585

**Published:** 2023-02-11

**Authors:** Vadim Chechekhin, Anastasia Ivanova, Konstantin Kulebyakin, Veronika Sysoeva, Daria Naida, Mikhail Arbatsky, Nataliya Basalova, Maxim Karagyaur, Mariya Skryabina, Anastasia Efimenko, Olga Grigorieva, Natalia Kalinina, Vsevolod Tkachuk, Pyotr Tyurin-Kuzmin

**Affiliations:** 1Department of Biochemistry and Regenerative Medicine, Faculty of Medicine, Lomonosov Moscow State University, 119991 Moscow, Russia; 2Institute for Regenerative Medicine, Medical Research and Educational Center, Lomonosov Moscow State University, 119991 Moscow, Russia; 3Burdenko Main Military Clinical Hospital, 105094 Moscow, Russia

**Keywords:** obesity, hypertension, sympathetic nervous system, MSCs, adrenergic receptors, contractile phenotype

## Abstract

Hypertension is a major risk factor for cardiovascular diseases, such as strokes and myocardial infarctions. Nearly 70% of hypertension onsets in adults can be attributed to obesity, primarily due to sympathetic overdrive and the dysregulated renin-angiotensin system. Sympathetic overdrive increases vasoconstriction via α1-adrenoceptor activation on vascular cells. Despite the fact that a sympathetic outflow increases in individuals with obesity, as a rule, there is a cohort of patients with obesity who do not develop hypertension. In this study, we investigated how adrenoceptors’ expression and functioning in adipose tissue are affected by obesity-driven hypertension. Here, we demonstrated that α1A is a predominant isoform of α1-adrenoceptors expressed in the adipose tissue of patients with obesity, specifically by multipotent mesenchymal stromal cells (MSCs). These cells respond to prolonged exposure to noradrenaline in the model of sympathetic overdrive through the elevation of α1A-adrenoceptor expression and signaling. The extent of MSCs’ response to noradrenaline correlates with a patient’s arterial hypertension. scRNAseq analysis revealed that in the model of sympathetic overdrive, the subpopulation of MSCs with contractile phenotype expanded significantly. Elevated α1A-adrenoceptor expression is triggered specifically by beta3-adrenoceptors. These data define a novel pathophysiological mechanism of obesity-driven hypertension by which noradrenaline targets MSCs to increase microvessel constrictor responsivity.

## 1. Introduction

Hypertension is a major risk factor for cardiovascular disease, particularly strokes and myocardial infarctions [1,2]. Increased vascular constriction is a result of the response of smooth muscle cells and pericytes to the imbalanced action of vasoconstrictors and vasodilators [3,4]. The sympathetic nervous system contributes to both arterial hypertension and associated vasoconstriction by activating alpha-adrenoceptors. In particular, the contraction of small vessels is controlled by the α1A-adrenoceptor isoform, which is predominantly expressed in human adipose tissue arteries [5]. Increased blood pressure is associated with excessive weight gain very frequently. Nearly 70% of hypertension onsets in adults can be attributed to obesity, primarily due to increased sympathetic nervous outflows and the dysregulated renin–angiotensin system [6]. In this study, we investigated how adrenoceptor expression and functioning in adipose tissues were affected by obesity-driven hypertension. Here, we demonstrate that α1A is a predominant isoform of α1-adrenoceptors expressed in adipose tissue of patients with obesity, specifically by multipotent mesenchymal stromal cells (MSCs). These cells are closely associated with adipose vasculature and partially overlap with pericytes [7]. In this study we demonstrate that MSCs enhance the sensitivity of blood vessels to noradrenaline and thus amplify post-synaptically the obesity-associated increase in sympathetic activity.

Noradrenaline elevates the sensitivity of MSCs to noradrenaline by activating the expression of α1A-adrenoceptors. We found a strong correlation between the prominence of the MSCs’ sensitization ability to noradrenaline and the donor’s arterial hypertension. Furthermore, MSCs acquire a contractile phenotype after sympathetic overdrive followed by noradrenaline treatment. We deciphered the molecular mechanism responsible for the up-regulation of α1A-adrenoceptors and revealed that the activation of the β3-adrenoceptor causes the sensitization of MSCs. These data suggest that MSCs, associated with the adipose vasculature, respond to obesity-associated elevations of noradrenaline by up-regulating α1A-adrenoceptors and further augmenting microvessel constrictor responses to elevated sympathetic outflow.

## 2. Materials and Methods

### 2.1. MSCs Isolation and Culturing

MSCs were isolated from the stromal-vascular fraction of subcutaneous adipose tissues of 36 donors (age 42.17 ± 2.07 years; BMI 29.36 ± 1.01 kg/m^2^, for details, see Appendix A) using enzymatic digestion [8]. These cells were CD45-/CD73+/CD105+/CD90+/NG2+/PDGFRβ+ [9]. All donors gave their informed consent. The study was conducted according to the guidelines of the Declaration of Helsinki and human primary cell collection was approved by the Local Ethic Committees of Burdenko Main Military Clinical Hospital (Moscow, Russia) and the Medical Research and Education Center of Lomonosov Moscow State University (IRB00010587, Moscow, Russia) approved the study protocol (#160, 22 July 2019 and #4, 4 June 2018, respectively). Donors medication presented in Appendix A. Primary MSCs were cultured in AdvanceSTEM Mesenchymal Stem Cell Media containing a 10% AdvanceSTEM Supplement (HyClone, Cytiva, Marlborough, MA, USA), 1% antibiotic–antimycotic solution (HyClone, Cytiva, Marlborough, MA, USA) at 37 °C in a 5% CO_2_ incubator (Binder, Tuttlingen, Germany, CB210). Cells were passaged at 70–80% confluency using Versen solution (Paneco, Moscow, Russia) and HyQTase solution (HyClone, Cytiva, Marlborough, MA, USA).

### 2.2. Immunofluorescent Detection of Alpha1A-Adrenoceptor

Part of the subcutaneous adipose tissue sample was placed in O.C.T. The compound (Sakura Inc., Tokyo, Japan) was frozen in liquid nitrogen for immunofluorescent analysis. α1A-adrenergic receptors were visualized by immunofluorescent staining of 10 μm frozen sections using antibodies against the α1A-(ab137123, Abcam, Cambrige, UK, 1:250), α1B-(ab169523, Abcam, Cambrige, UK, 1:100), α1D-(sc-390884, Santa Cruz, 1:100), β3-adrenergic receptors (H00000155-B01P, Abnova, Taipei City, Taiwan, 1:100). Briefly, sections were fixed in 4% paraformaldehyde for 10 min. After several washes, PBS sections were incubated in 0.1% BSA containing 10% normal donkey serum to block the non-specific binding of antibodies. This was followed by incubation with specific primary antibodies for 1 h and subsequent extensive washing in PBS. Then, sections were incubated with Alexa594-conjugated donkey anti-rabbit (Molecular Probes, Eugene, OR, USA). To assess the co-localization of cells expressing an alpha1A-adrenergic receptor and blood vessels, we performed double immunofluorescent staining: sections were incubated with a mix of antibodies against adrenergic receptors and antibodies to endothelial cells (CD31, IR610, Dako, Santa Clara, CA, USA) or smooth muscle cells (alpha-SMA, M0851, Dako, Santa Clara, CA, USA) or PDGFRβ (P7679, Sigma-Aldrich, St. Louis, MO, USA) or sympathetic nerves (tyrosine hydroxylase, AB152, Sigma-Aldrich, St. Louis, MO, USA). This was followed by incubation with the Alexa594-conjugated donkey anti-rabbit antibody (A-21207, Thermo Fisher Scientific, Waltham, MA, USA) and Alexa488-conjugated anti-mouse antibody (A-11001, Thermo Fisher Scientific). Cell nuclei were counterstained with DAPI (D9542, Sigma-Aldrich, St. Louis, MO, USA), and sections were mounted in Aqua Poly/Mount (Cat#18606, Polysciences Inc., Warrington, PA, USA). For the negative controls, mouse or rabbit non-specific IgGs were used in an appropriate concentration. Images were obtained using a confocal microscope LSM 780 and ZEN2010 software (Zeiss, Jena, Germany).

### 2.3. Western-Blotting

Cells were grown to 60% confluence in 100 mm Petri dishes with DMEM LG (Gibco, USA) containing 10% FBS (Gibco, USA) and 1% antibiotic–antimycotic solution (HyClone, Cytiva, Marlborough, MA, USA). The cells were stimulated with 10^−6^ M noradrenaline, incubated for 1 h (sympathetic overdrive model), and washed three times with a growth medium. After 1, 3, 6, or 24 h of stimulation, the growth medium was quickly removed, and the dishes were transferred on ice. The cells were rinsed with ice-cold PBS (5.2 mM Na_2_HPO_4_, pH 7.4, and 150 mM NaCl) and scraped by a rubber cell scraper (Corning) in 3× SDS buffer (6% SDS, 0.2 M Tris, pH 6.8, 40% glycerol and 0.03% bromophenol blue). The samples were passed 5 times through a 30-gauge needle to splinter DNA. ASC52telo cells, after knockout, were lysed using the same protocol. The total protein concentration was measured using Bio-Rad protein assay based on the Bradford dye-binding method (BioRad, Hercules, CA, USA).

Proteins were separated by Any kD Mini-PROTEAN TGX Stain-Free Protein Gels (BioRad, Hercules, CA, USA) in the Mini-PROTEAN 3 BioRad Units using a Tris/Glycine/SDS buffer (diluted from 10× premixed buffer, containing 25 mM Tris, 192 mM glycine, 0.1% SDS, pH 8.3, BioRad, Hercules, CA, USA). The loading amount was controlled by immunostaining for vinculin. The proteins were transferred onto 0.45 μm PVDF membranes (Amersham, Cytiva, Marlborough, MA, USA) for 1 h at 350 mA in the buffer containing 25 mM Tris, 0.192 M glycine, pH 8.3, 20% ethanol, 0.02% sodium lauryl sulfate. The membranes were blocked for 1 h in 5% non-fat milk in TBST (25 mM Tris/HCl, pH 7.4, 150 mM NaCl, 0.1% Tween-20). The membranes were incubated overnight at 4oC with an appropriate primary antibody in a dilution recommended by the supplier, washed 3 times in TBST, and incubated for 1 h at room temperature with the peroxidase-conjugated secondary antibodies. For alpha1A-adrenergic receptor analysis, we used a rabbit monoclonal antibody (Abcam, Cambrige, UK, ab137123). Anti-β2- (Abcam, Cambrige, UK, ab61778) and anti-β3-adrenoceptor (Acris, Herford, Germany, H00000155-B01P) antibodies were used to confirm the downregulation of expression in ASC52telo knockout. All blocking procedures, antibody incubation, and washings were carried out in TBST and supplemented with 5% dry nonfat milk. The protein bands were visualized by enhanced chemiluminescence (West Pico, USA) on the ChemiDoc Imaging System (BioRad, Hercules, CA, USA). Each membrane was developed for at least two different time intervals to ensure the linearity of the ECL signal. The quantitative analysis was achieved using ChemiDoc Imaging System (BioRad, Hercules, CA, USA) software.

### 2.4. Single-Cell Droplet-Based RNA-Seq Library Preparation and Sequencing

We used MSCs with a high level of sensitization ability after sympathetic overdrive. MSCs were treated by noradrenaline 6 h after sympathetic overdrive. The cells were analyzed 12 h after noradrenaline (Sympathetic overdrive—Nor). The control cells were treated by the vehicle two times (Control). The single-cell suspensions of MSCs were converted to barcoded scRNA-seq libraries using the Chromium Next GEM Single Cell 3’ GEM (10× Genomics, Pleasanton, CA, USA), aiming for 10,000 cells per library. Samples were processed using Library & Gel Bead Kit v3.1 barcoding chemistry (10× Genomics, Pleasanton, CA, USA). Single samples were processed in a single well of a PCR plate, allowing all cells from a sample to be treated with the same master mix and in the same reaction vessel. Samples (Control and Sympathetic overdrive—Nor) were processed in parallel in the same thermal cycler and Illumina HiSeq1500 sequencing system.

### 2.5. Collagen Gel Contraction Assay

A collagen gel contraction assay was performed as described previously [10]. Briefly, cells were trypsinized and centrifuged. A total of 100,000 cells per well in DMEM were added to the collagen (PC11-NCL, Imtek, Nashua, NH, USA) and neutralized with 1M NaOH (3 mg/mL in 0.1% acetic acid, Imtek, Nashua, NH, USA). The matrix was allowed to polymerize at RT for 30 min. Then, an advanced stem medium with a 10% supplement was added, and the matrix disk was detached from the well walls to allow for contraction. Noradrenaline was added to the culture medium 30 min after detaching the model sympathetic overdrive; 1 h later, the disks were washed off with DMEM LG, and the medium was changed to fresh Advance stem with 10% supplement. Noradrenaline was added again to the culture medium 6 h after adding the first stimulus. The matrix disk area was measured before the second stimuli and 2, 12, 24, 28 h after using Nikon SMZ18. % matrix contraction was calculated as (area of the shortened disk)/(area of the disk before stimulus)×100.

### 2.6. MSCs Treatment and Ca^2+^ Imaging

Adrenergic receptor activation was assessed using noradrenaline (Abcam, Cambrige, UK, ab120717, 1 µM) and Ca^2+^ imaging. Cells were grown at low density to prevent cell-to-cell communication during the calcium imaging. To analyze the amount of functionally active alpha1A-adrenergic receptors after sympathetic overdrive modeling, cells were seeded in 24 well plates at a low density, and we stimulated cells with noradrenaline for 1 h, washed them three times using Hanks solution, and incubated the cells in a full growth medium for an additional 4 h. The cells were loaded with Fluo-8 (Abcam, Cambrige, UK, ab142773, 4 µM) in Hanks solution with 20 mM Hepes for 1 h before the experiment. To measure the percentage of the responded cells, we recorded the baseline for 5 min, then once added noradrenaline. Ca^2+^ transients were measured in individual cells using an inverted fluorescent microscope Nikon Eclipse Ti equipped with an objective CFI Plan Fluor DLL 10×/0.3 (Nikon, Tokyo, Japan) and with a digital EMCCD camera Andor iXon 897 (Andor Technology, Belfast, UK). We used simultaneous measuring of 6 × 6 fields of view in the large image mode to increase the number of analyzed cells. Movies were analyzed using NIS-Elements (Nikon) and ImageJ software. Alterations of cytosolic Ca^2+^ were quantified by relative changes in the intensity of Fluo-8 fluorescence in individual cells. The percentage of responded cells was measured as the ratio of the number of responded cells to the number of all the analyzed cells. To evaluate what isoforms of β-adrenergic receptors regulated the number of MSCs responding to noradrenaline, we added to cells that were either β1-antagonist CGP20712 (Tocris, Bristol, UK, Cat#1024, 100 nM) or β2-antagonist ICI118551 (Tocris, Bristol, UK, Cat#0821, 50 nM) or β3-antagonist SR59230A (Tocris, Bristol, UK, Cat#1511, 250 nM) with noradrenaline in the modeling of sympathetic overdrive.

### 2.7. CRISPR/Cas9-Mediated Knockout of ADRB2/ADRB3 Genes in ASC52telo Cell Line

A pair of lentivirus constructs was used to knockout *ADRB2*/*ADRB3* genes. Lentiv2-AncBE4maxR33A-SpCas9D10A(NG)-P2A-GFP, for the expression of SpCas9D10A(NG)-based highly specific cytosine deaminase and GFP, and Lentiv2-gRNA-RFP for the expression of gRNA and RFP (Appendix A), were assembled in our laboratory using Addgene vectors LentiCRISPRv2GFP (#82416), pCMV_AncBE4max_P2A_GFP (#112100), pX459-SpCas9-NG (#171370). gRNA protospacers were cloned in Lentiv2-gRNA-RFP using BsmBI sites. The sequences of gRNA protospacers used were *ADRB2*-5′-GTACCAGAGCCTGCTGACCA and *ADRB3*-5′-ACTCCAGACCATGACCAACG. Lentiviral particles encoding components of a CRISPR/Cas9 genome editing system were assembled as described before [11]. ASC52telo transduction was performed as described earlier [12]. To confirm *ADRB2*/*ADRB3* knockout, genomic DNA was isolated from ASC52telo populations and amplified using the primers listed in Table 1. Amplicons were sequenced by Sanger sequencing, and the results were analyzed using Chromas 2.6.6 software (Technelysium Pty Ltd., South Brisbane, Australia).

### 2.8. Quantification and Statistical Analysis

#### 2.8.1. Analysis and Quality Control of Single-Cell RNA-Seq Data

Samples were mapped to the reference genome (human reference genome NCBI build 38, GRCh38) using CellRanger 6.1.2 (10× Genomics, Pleasanton, CA, USA). We used the following quality control criteria: cells with <2500 or >7500 detected genes or <7000 or >70,000 RNA counts or over 5% unique molecular identifiers (UMIs) derived from the mitochondrial genome were filtered out as low-quality cells. Data from samples were processed using R-studio 1.4 with R 4.1.2 and Seurat 4.0.4, regressing out mitochondrial genes [13]. The integration of datasets was performed using the Seurat function IntegrateData. The principal component analysis of integrated datasets was performed on the variable genes, and 20 principal components were used for cell clustering (resolution = 0.3) and UMAP dimensional reduction. The cluster markers were found using the FindAllMarkers function. Cell types were manually annotated based on the cluster markers using g:GOSt functional profiling. To perform RNA velocity analysis, we estimated unspliced and spliced mRNA counts using velocity 0.17.16 [14]. The resulting loom files were merged with integrated datasets and then analyzed with the scVelo package 0.2.4 [15]. All scVelo functions were used with default parameters. A trajectory inference was performed with the Dynverse collection of R packages using the Paga-tree method [16,17]. A sample of CD45 negative cells of the stromal vascular fraction—are represented by SRR12423012 (GSE155960) [18].

#### 2.8.2. Data Representation and Statistical Analysis

Statistical analysis was performed using SigmaPlot 12.5 software (Systat Software Inc., San Jose, CA, USA). Data were assessed for normality of distribution using the Shapiro–Wilk test. Values are expressed as the mean ± standard error of the mean (SEM). A comparison of two independent groups was performed by Mann–Whitney U-criteria (M-U test) for non-normal distributed data. Multiple comparisons were made using the Kruskal–Wallis test (one-way ANOVA on ranks) with the subsequent application of Dunn criteria. Statistical significance was defined as *p*-value < 0.05.

## 3. Results

### 3.1. α1A-Adrenoceptor Is a Predominant Adrenoceptor Isoform Expressed by MSCs in Adipose Tissue Vessels

The analysis of the spatial distribution of α1-adrenoceptor isoforms in the adipose tissue of normotensive donors with obesity and hypertensive donors with obesity has revealed that the α1A-adrenoceptor is a predominant one. In normotensive patients with obesity, α1A-adrenoceptor was found in cells adjacent to the CD31-labeled endothelium in blood vessels (Figure 1A and Appendix A). These cells did not contain alpha-smooth muscle actin and, therefore, should not be considered smooth muscle cells (SMCs) (Figure 1B and Appendix A). Cells that are positive for α1A-adrenoceptor are located between the endothelium and SMC. All these cells express PDGFR-β, which is a commonly used marker of MSCs (Figure 1C and Appendix A) [9]. MSCs are frequently located in close proximity to structures containing tyrosine-hydroxylase, which indicates that these cells are likely to be directly innervated by sympathetic fibers (Appendix A). The expression of α1A-adrenoceptor was markedly elevated in the adipose tissue of hypertensive donors with obesity. Here, α1A-adrenoceptors were expressed predominantly by MSCs but were also found in some α-SMA expressing cells comprising adipose vasculature (Figure 1D–H and Appendix A). Other α1-adrenoceptor isoforms (α1B and α1D) were found less frequently. α1D-adrenoceptor was found in stromal cells within adipose tissue interstitium but not within vascular cells (Appendix A). There was no difference between males and females in percentage of perivascular cells expressing alpha1A-adrenoceptor and in share of perivascular cells with alpha1A-adrenoceptor and alpha-smooth muscle actin colocalization (Appendix A).

### 3.2. MSCs Sensitivity to Noradrenaline after Sympathetic Overdrive Correlates with Elevated Blood Pressure in Patients with Obesity

To determine if a pathological increase in the sympathetic nervous outflow, so-called sympathetic overdrive contributes to the elevation of α1A-adrenoceptor expression, we isolated MSCs from the adipose tissue of normotensive and hypertensive donors. Prolonged exposure to noradrenaline caused the transitory elevation of α1A-adrenoceptor expression in MSCs (Figure 2A). Functional testing using a single-cell Ca^2+^ influx assay showed that the portion of cells responding to noradrenaline increased in this model of sympathetic overdrive (Figure 2B,C, Appendix A). Clinical data analysis indicated that the extent of this effect correlated with the systolic and mean blood pressure (Figure 2D, Appendix A). Furthermore, MSCs from hypertensive patients with obesity readily demonstrated such a response, whereas cells from normotensive patients with obesity did not (Figure 2E). There was no correlation between blood pressure and the value of MSCs sensitization to noradrenaline after sympathetic overdrive modeling in patients with BMI < 30 kg/m^2^ (Appendix A). Also, there was no correlation between sex, BMI or age and the value of MSCs sensitization to noradrenaline after sympathetic overdrive modeling in patients (Appendix A).

### 3.3. Single Cell RNA-Seq Reveals MSCs Phenotype Shift towards Contractile Cells

The expression of α1A-adrenoceptor points to MSCs as a central target of sympathetic overdrive. These cells regulate key aspects of adipose tissue growth and functioning, such as adipocyte generation, vascularization, and ECM turnover. Therefore, we have analyzed the potential mechanisms of MSCs involved in the development of obesity-driven hypertension. We have performed a single-cell RNA sequencing (scRNAseq) of MSCs treated by noradrenaline after sympathetic overdrive modeling using vehicle-treated cells as a control (Figure 3A). We integrated control and experimental datasets and identified several functional clusters, including fibroblasts-like cells (Cluster 4: *VIM*, *COL1A1*, *COL5A1*, laminins, *FN1*, *DCN*, *FBLN1*, *FBLN2*, *MFAP5*, and extracellular matrix-associated proteins *LOXL1*, *LUM*, *MMP2* enriched); mitotic cells (Cluster 0: genes associated with G2, M, and S phases of the cell cycle), and contractile cells (Cluster 5: *ACTG2*, *ACTA2*, *MYL9*, *MYH11* and regulators *MYLK*, actin-associated proteins *TAGLN*, *LMOD1*, *SYNPO2*, *CALD1*, and *CNN1*, adhesion molecules *ITGA7*, *ESAM*, *MCAM*, *THBS1* as well as *PDGFA* and *VEGFA*) (Figure 2C, Appendix A, Appendix A) [19]. In the experimental group, the presence of fibroblast-like cells comprising Cluster 4 showed a 1.7 times increase. The most prominent rise was observed for contractile cells (Cluster 5), which increased up to four times.

In order to define the precise phenotype of Cluster 5 contractile cells, we integrated our datasets with a published stromal-vascular fraction dataset of a patient with obesity. In the resulting integrated dataset, Cluster 5 contractile cells aligned exclusively with smooth muscle cells. In light of these data, we hypothesized that sympathetic overdrive followed by noradrenaline stimulation caused a phenotype shift towards smooth muscle cells (Appendix A). Moreover, the transcription dynamic analysis by Velocyto and scVelo revealed common trends toward contractility increase. Thus, the transcription of unspliced (immature) mRNAs associated with contraction increased in most of the cells (Figure 2C, Appendix A). Trajectory inference analysis showed that Cluster 5 cells likely arose from Cluster 3 and 2 (synthetic cells), indicating their differentiation rather than proliferation (Appendix A). These data indicate that MSCs exposed to sympathetic overdrive acquire a contractile smooth muscle-like phenotype after noradrenaline treatment.

To test how MSCs’ exposure to sympathetic overdrive affects their contractility at the functional level, we used the collagen disk retraction assay (Figure 3D–F, Appendix A). MSCs were polymerized into a 3D-collagen matrix and stimulated with noradrenaline or a vehicle according to our model of sympathetic overdrive (see also Appendix A). The collagen disc area was measured 30 min and 24 h after the second noradrenaline treatment. Although a single addition of noradrenaline had no immediate effect on the collagen discs area (Figure 3E, Appendix A), we observed a significant shrinkage in collagen disks containing cells stimulated by noradrenaline after the sympathetic overdrive modeling (“SO/Nor” at Figure 3D), compared to disks with untreated cells (Figure 3F, Appendix A).

### 3.4. β3-Adrenoceptor Is Responsible for α1A-Adrenoceptor Elevation upon Sympathetic Overdrive

It is known that α1A-adrenoceptor expression is up-regulated by β-adrenoceptors [20]. To unveil the precise mechanism of α1A-adrenoceptor elevation, we exposed MSCs to sympathetic overdrive in the presence of selective beta-adrenoceptor inhibitors. The β3-antagonist fully blocked the effect of sympathetic overdrive, whereas the β2-antagonist only insignificantly reduced it, and the beta 1 antagonist had no effect (Figure 4A). Using the CRISPR/Cas9 knockout of β2- or β3-adrenoceptors in MSCs, we have also demonstrated that exposure to the sympathetic overdrive up-regulated α1A-adrenoceptor in MSCs with β2-knockout but not in cells lacking β3-adrenoceptors (Figure 4B,C, Appendix A). Taken together, these data suggest that sympathetic overdrive elevates α1A-adrenoceptor expression in MSCs, specifically via β3-adrenoceptors. Immunofluorescent analysis of β3-adrenoceptor distribution in the adipose tissue of normotensive and hypertensive patients with obesity has revealed that this adrenoceptor is localized in MSCs (Figure 4D,E).

## 4. Discussion

This study provides novel evidence that MSCs within adipose tissue vasculature may be critically important in promoting life-threatening obesity complications, particularly obesity-induced arterial hypertension. These cells are responsible for adipose tissue remodeling since they generate new adipocytes and myofibroblasts, as well as facilitate the blood supply and innervation of growing adipose tissue [21,22]. In situ, MSCs often reside in close proximity to endothelial cells and, therefore, are called “pericytes” due to such a location. The sympathetic innervation of pericytes and smooth muscle cells is necessary for blood pressure control [23]. Not all patients with obesity are diagnosed with hypertension despite obesity which is accompanied by sympathetic overdrive as a rule. In this study, we have demonstrated that α1A is a predominant α-adrenoceptor isoform in adipose tissue vasculature. The elevated expression of α1A-adrenoceptor and increased MSCs sensitivity to noradrenaline could comprise the pathological mechanism of an individual’s susceptibility to obesity-driven hypertension development [24,25]. Thus, MSCs within adipose tissue respond to obesity-associated sympathetic overdrive through the up-regulation of the α1A-adrenoceptor, which correlates well with the level of mean and systolic blood pressure of patients with obesity.

According to our single-cell RNAseq data, an increased α1A-adrenoceptor expression and signaling could facilitate the acquisition of a smooth muscle-like contractile phenotype by MSCs. These data indicate that sympathetic overdrive does not affect the MSCs’ contractility per se but has a permissive effect on their further contractile response to noradrenaline. Transcriptome analysis is in line with the in situ distribution of the α1A-adrenoceptor in adipose tissue of patients with obesity. In normotensive patients with obesity, α1A-adrenoceptor is specifically expressed by MSCs, whereas in hypertensives, we also found alpha-smooth muscle actin positive cells containing this adrenoceptor. This allows us to hypothesize that cells co-expressing alpha-smooth muscle actin and α1A-adrenoceptor arise from a shift in the MSCs’ phenotype. Taken together, these data illuminate a possible mechanism of origin for α1A-adrenoceptor positive smooth muscle-like cells in hypertensive patients with obesity.

Sympathetic overdrive followed by noradrenaline treatment causes an increase in the number of cells expressing ECM (Cluster 4). These data suggest that sympathetic overdrive can contribute to hypertension development by promoting extracellular matrix overproduction. This, in turn, is necessary for the development of vascular stiffness. Increased vascular stiffness may arise both from extracellular matrix overproduction and the actin-dependent augmentation of the intracellular stiffness of vascular cells. Our scRNAseq data indicate that intracellular stiffness per se was not affected in the experimental group since the expression of genes associated with intracellular stiffness (*ACTA2*, *ITGB1*, *FN1*) did not change [26]. The increased stiffness of adipose tissue vasculature creates a positive-feedback loop for further obesity progression as well as obesity-associated hypertension [27]. These data are consistent with the observation that the excessive activation of α1A-adrenoceptors in the vasculature is linked to the development of vascular stiffness and points to MSCs as a cellular target [28].

Our data point to β3-adrenoceptor activation as a main trigger of α1A-adrenoceptor elevation in MSCs, which is consistent with the primary role of the β3-adrenoceptor in the sympathetic regulation of MSC activity observed in the bone marrow [29]. Furthermore, β3-adrenoceptor polymorphism contributes to the onset and maintenance of hypertension [30]. Since this adrenoceptor is a recognized therapeutic target [31,32], the potential effects of its antagonists on obesity-associated hypertension should be taken into consideration.

## 5. Conclusions

The disclosure of the molecular mechanisms for varied sensitivity to sympathetic overdrive is essential for selecting the methods of prevention and treatment of obesity-associated arterial hypertension. Our data suggest that MSCs within adipose tissue may respond to the obesity-associated elevation of noradrenaline via the up-regulation of the α1A-adrenoceptor. Increased α1A-adrenoceptor could further augment microvessel constrictor responses to elevated sympathetic outflow. Increased α1A-adrenoceptor signaling on MSCs suggests a new mechanism of simultaneous alpha- and beta-blockers effectiveness for arterial hypertension treatment in patients with obesity [33]. This study provides novel evidence that MSCs within adipose tissue vasculature may be critically important in promoting life-threatening obesity complications, particularly obesity-induced arterial hypertension.

## Figures and Tables

**Figure 1 cells-12-00585-f001:**
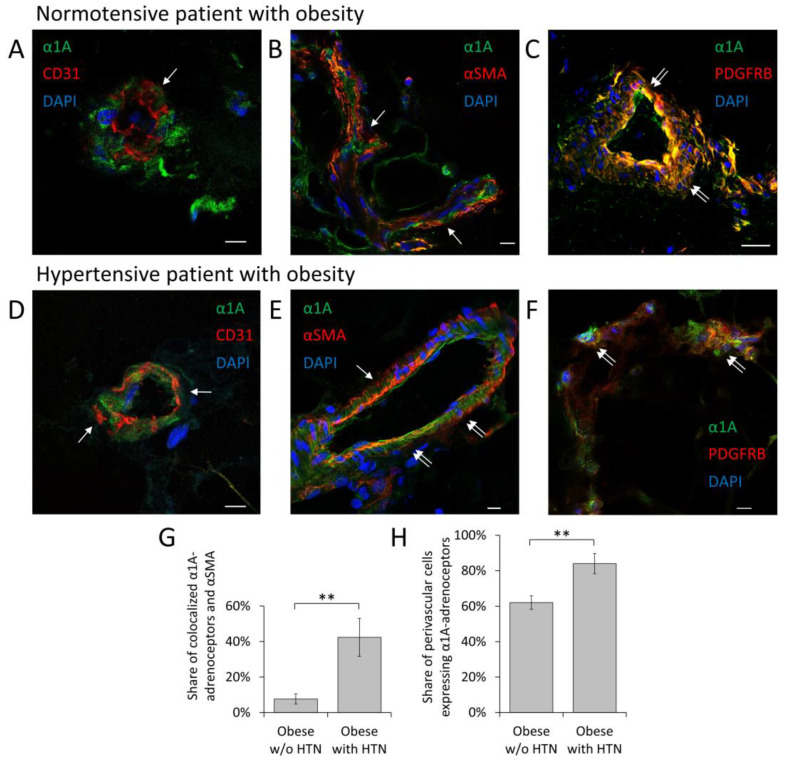
Alpha1A-adrenoceptor expressing MSCs have perivascular distribution. (**A**–**F**) Confocal images of frozen sections of human subcutaneous adipose tissue. (**A**–**C**) Adipose tissue of normotensive patients with obesity. (**D**–**F**) Adipose tissue of hypertensive patients with obesity. (**A**,**D**) Alpha1A-adrenoceptor (green) is localized outside of CD31 positive endothelial cells (red, single arrow). (**B**) Alpha1A-adrenoceptor (green) is localized outside of alpha-smooth muscle actin positive cells (red, single arrow). (**E**) Alpha1A-adrenoceptor (green) is localized in the portion of alpha-smooth muscle actin positive cells (yellow, double arrow). (**C**,**F**) Alpha1A-adrenoceptor (green) is localized in MSCs (PDGFRβ—green) (yellow, double arrow). Nuclei stained with Dapi (blue). Scale bar 10 μm. (**G**) In hypertensive patients with obesity, alpha1A-adrenoceptor colocalizes with alpha-smooth muscle actin in some perivascular cells. Share of perivascular cells with alpha1A-adrenoceptor and alpha-smooth muscle actin colocalization calculated based on immunofluorescent staining. (**H**) Percentage of perivascular cells expressing alpha1A-adrenoceptor in normotensive patients with obesity and hypertensive patients with obesity. Mean ± SEM, *n* = 8–23, ** *p* < 0.01 (Mann–Whitney U Test).

**Figure 2 cells-12-00585-f002:**
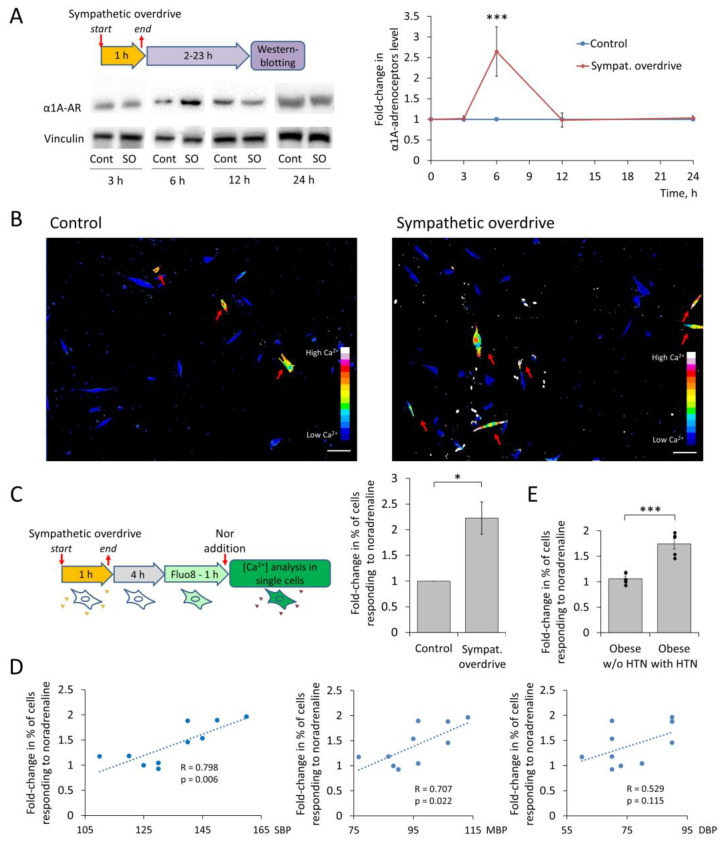
Sympathetic overdrive modeling up-regulates alpha1-adrenoceptor expression and signaling on MSCs. (**A**) Western blot analysis of α1A-adrenergic receptors after modeling of sympathetic overdrive, representative images, and quantification of bands intensity, *n* = 5–11, *** *p* < 0.001 (Mann–Whitney U Test). (**B**) Ca^2+^ registration after noradrenaline addition in single MSCs using Fluo-8 dye from the same cell preparation, in vehicle-treated cells and after sympathetic overdrive modeling. Scale bar 100 μm. (**C**) Scheme of intracellular Ca^2+^ increase registration in MSCs after sympathetic overdrive modeling followed by noradrenaline addition. Cells were treated by noradrenaline (Sympathetic overdrive) or a vehicle (Control) for 1 h and were subjected to noradrenaline treatment together with calcium imaging. Fold change in the share of MSCs responded to noradrenaline 6 h after sympathetic overdrive, mean ± SEM, *n* = 9–23, * *p* < 0.05 (Mann–Whitney U Test). (**D**) Correlation between systolic, diastolic, and mean blood pressure and the magnitude of MSCs response to noradrenaline after sympathetic overdrive. (**E**) Fold-change in share of MSCs from donors with obesity with or without hypertension responding to noradrenaline 6 h after sympathetic overdrive, mean ± SEM, *n* = 5–7, *** *p* < 0.001 (Mann–Whitney U Test). HTN—hypertension, SBP—systolic blood pressure, DBP—diastolic blood pressure, MBP—mean blood pressure, SO—sympathetic overdrive, Nor—Noradrenaline.

**Figure 3 cells-12-00585-f003:**
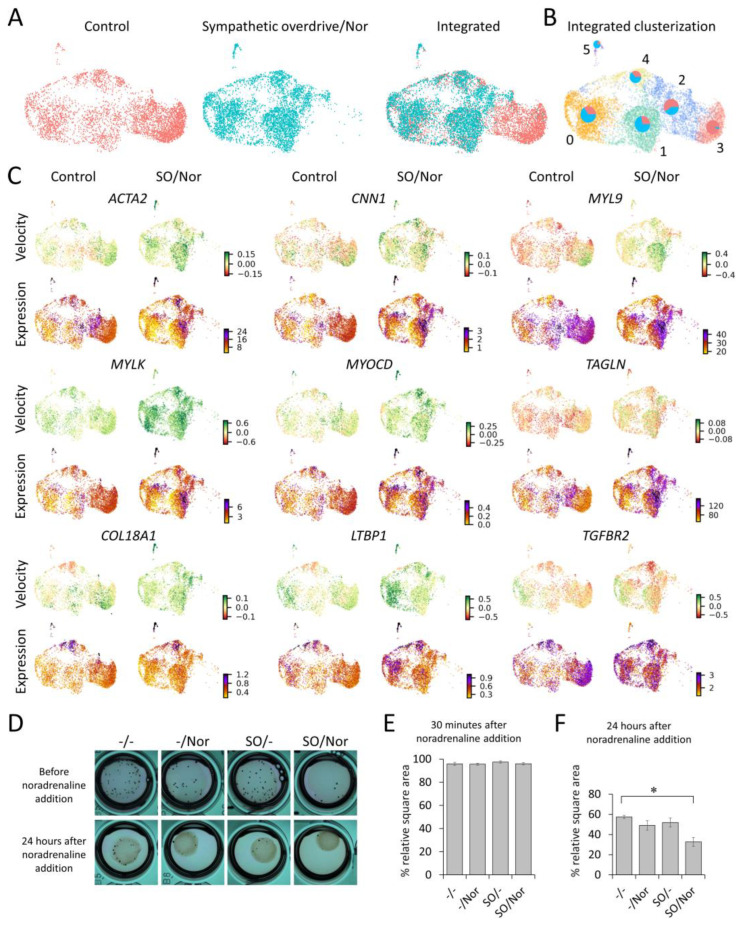
Evaluation and functional analysis of MSCs’ contractile properties after sympathetic overdrive followed by noradrenaline. (**A**–**C**) scRNAseq analysis of MSCs after sympathetic overdrive followed by noradrenaline addition. (**A**) Uniform manifold approximation and projection (UMAP) representation of MSCs from the control and experimental sample. Samples are positioned in the same coordinates by their transcriptomic similarity separately and after integration. Cells were integrated and visualized by Seurat. (**B**) Clustering of integrated samples with a visualized proportion of cells in each cluster. Red sectors of circles correspond to the portion of control cells in a particular cluster, blue sectors correspond to the cells after sympathetic overdrive followed by noradrenaline addition. (**C**) mRNA velocities (velocity) and expression (expression) of some particular genes projected onto UMAP plots of control and experimental samples. (**D**) Collagen disks with MSCs’ contraction just before noradrenaline addition and 24 h after noradrenaline addition. Vehicle-treated MSCs (-/-), MSCs treated with noradrenaline alone (-/Nor); MSCs after sympathetic overdrive without followed noradrenaline treatment (SO/-); MSCs treated by noradrenaline after sympathetic overdrive (SO/Nor) (*n* = 3–6). (**E**,**F**) Relative square area of collagen disks containing MSCs 30 min (**E**) and 24 h (**F**) after noradrenaline treatment. * *p* < 0.05 calculated with Kruskal–Wallis One Way ANOVA on Ranks.

**Figure 4 cells-12-00585-f004:**
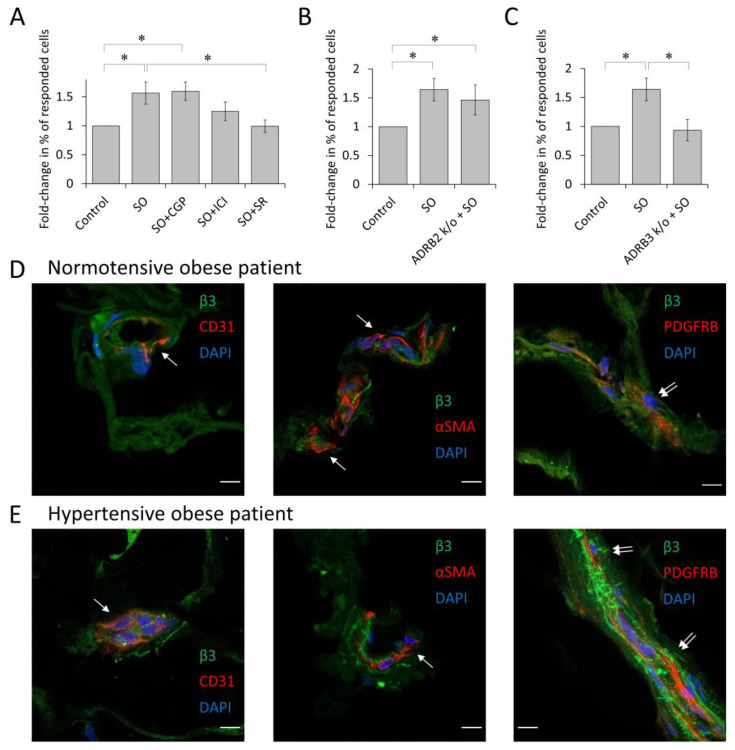
β3-adrenergic receptors are responsible for alpha1A-adrenoceptors up-regulation in MSCs. (**A**) Fold-change in the percentage of MSCs responded to noradrenaline after sympathetic overdrive (SO) or sympathetic overdrive in the presence of the β1-antagonist CGP20712 (100 nM, CGP), β2-antagonist ICI118551 (50 nM, ICI), or β3-antagonist SR59230A (250 nM, SR) (*n* = 12–19). (**B**,**C**) Fold-change in the percentage of cells that responded to noradrenaline after sympathetic overdrive of either control ASC52telo cells (SO) or cells after ADRB2 (**B**) or ADRB3 (**C**) knockout (*n* = 4–7). (**D**,**E**) Confocal images of frozen sections of human subcutaneous adipose tissue. β3-adrenoceptor localized in PDGFRβ positive MSCs in vessels of subcutaneous adipose tissue of normotensive patients with obesity (**D**) and hypertensive (**E**) patients with obesity (double arrow). Cells expressing β3-adrenergic receptor (green), CD31 positive endothelial cells (red), alpha smooth muscle actin positive cells (red), and PDGFRβ positive MSCs (red). Nuclei stained with Dapi (blue). Scale bar 10 μm. * *p* < 0.05 calculated with Kruskal–Wallis and one way ANOVA on ranks.

**Table 1 cells-12-00585-t001:** Primers, used for the amplification of the target DNA site within the range of editing.

Name	Sequence	Amplicon Length, bp	T_melting_, °C
h*ADRB2*-test-f	GCAACTTCTGGTGCGAGTTT	415	59.5
h*ADRB2*-test-r	AAGCGGCCCTCAGATTTGTC		
h*ADRB3*-test2-f	GCAGTAGATGAGCGGGTTGAA	830	60
h*ADRB3*-test2-r	ACGTGTTCGTGACTTCGCT		

## Data Availability

Original Western blot images have been deposited at Mendeley and are publicly available as of the date of publication (doi: 10.17632/sfrbyc2s9y.1). Microscopy and single-cell RNA-seq data reported in this paper will be shared by the lead contact upon request.

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
