# Peer review of "Alpha1A- and Beta3-Adrenoceptors Interplay in Adipose Multipotent Mesenchymal Stromal Cells: A Novel Mechanism of Obesity-Driven Hypertension"

_cells, 2023, doi:10.3390/cells12040585_

Round 1

Reviewer 1 Report

In this paper the authors propose a mechanism by which MSC could be involved in obesity-driven hypertension.

major point

However the authors did not include as controls:  hypertensive non-obese patients. Indeed, it seems important to know whether the increased sensitivity to NA, could be  due to the hypertensive state,  independent on obesity, or not .

In figure 1, the authors indicate that obese AT express more alpha-1 adrenoreceptors than controls. However, this is not clearly shown. Could the authors show quantitative data ? Moreover, authors indicate that the alpha-1 adrenoreceptors are predominantly expressed by MSCs , but there is no labeling of MSCs. Thus the authors should double label MSCs with one of its marker ( such as CD73) plus alpha1-adrenergic receptor. In this figure  is missing a control from hypertensive non obese adipose tissue 

In figure 2A, the authors indicated that they used MSCs, but they did not indicate whether those cells were from obese and hypertensive patients or not. In this figure, there is no comparison between groups ( obese, and hypertensive or obese and  normotensive),

Figure 3: In this figure, it is not clear whether RNAseqs were performed in patients with hypertension or not. And here again is missing a comparison between groups, and  the hypertensive non obese  group

Minor points

In figure 3C, are missing legends  to indicate control and SO

Author Response

Response to Reviewer 1 Comments

Major point

1) However the authors did not include as controls:  hypertensive non-obese patients. Indeed, it seems important to know whether the increased sensitivity to NA, could be  due to the hypertensive state,  independent on obesity, or not.

Response 1: Arterial hypertension is a multifactorial disease, and in non-obese patients there can be many different causes of the development of the disease. In contrast, in most obese patients, the etiology of hypertension is rather common, as described, for example, in (https://doi.org/10.1161/CIRCRESAHA.116.305697 Circulation Research. 2015;116:991–1006). Thus, we suggest that non-obese hypertensive patients may cause unnecessary confusion at work, and that is not an appropriate control. That suggestion is also confirmed by the absence of any signs of correlation between the ability to sensitization of a1A-adrenoceptors and arterial hypertension in non-obese patients (see the table and graphs below). We have added these results in Supplement. Thus we suggest that an ability to sensitize a1A-adrenoceptors is a risk factor for the development of hypertension, associated with obesity but not in a lean state.

2) In figure 1, the authors indicate that obese AT express more alpha-1 adrenoreceptors than controls. However, this is not clearly shown. Could the authors show quantitative data ? Moreover, authors indicate that the alpha-1 adrenoreceptors are predominantly expressed by MSCs , but there is no labeling of MSCs. Thus the authors should double label MSCs with one of its marker ( such as CD73) plus alpha1-adrenergic receptor. In this figure  is missing a control from hypertensive non obese adipose tissue. 

Response 2: We have added the results of immunofluorescent images processing in Figure 1 (G and H). We detected MSCs in vessels by staining with PDGFRbeta, which is a common MSC marker (https://doi.org/10.1186/s13287-015-0209-8). Colocalization of a1-adrenoceptors and MSCs was assessed by co-staining of the receptor and PDGFRbeta. We have clarified this point in the text. Also we added arrows that show colocalization of alpha1A-adrenergic receptors and PDGFRbeta positive MSC into Figure 1. 

3) In figure 2A, the authors indicated that they used MSCs, but they did not indicate whether those cells were from obese and hypertensive patients or not. In this figure, there is no comparison between groups ( obese, and hypertensive or obese and  normotensive).

Response 3: In Figure 2, there is no comparison between hypertensive and normotensive patients. Here, we analyzed the mechanisms of MSCs sensitization to noradrenaline after sympathetic overdrive modeling. For this purpose, we first examined whether the MSCs of a particular donor increased the number of responding cells after sympathetic overdrive modeling. And then we carried out works of Figure 2 only on cells capable of sensitizing a1A-adrenergic receptors. The donors of both groups, obese and lean, were included into the statistics of experiments presented in Figure 2. Western-Blotting results are available in the Mendeley database (see Data availability statement).

4) Figure 3: In this figure, it is not clear whether RNAseqs were performed in patients with hypertension or not. And here again is missing a comparison between groups, and  the hypertensive non obese  group

Response 4: We used the scRNAseq methods to elucidate how cell exposure to excessive action of noradrenaline affects the transcriptome of MSCs at single cell level. As well as for the implementation of experiments Fig. 2, we chose a donor whose MSCs were capable of sensitizing to noradrenaline. That was a donor with obesity capable of sensitizing to noradrenaline. 

Minor points

5) In figure 3C, are missing legends  to indicate control and SO

Response 5: We have added marks Control and SO/Nor in Figure 3C.

Reviewer 2 Report

The aim of this study was to investigate how adrenoceptors expression and functioning in adipose tissue is affected by obesity-driven hypertension.

This study is interesting; however, I have some comments:

First, is better to use People-First Language, i.e. "individuals with obesity" rather than "obese individuals", generally, 'obese' should not be used as an adjective.

Material and Methods section

In table 1S, please change gender by sex (female or male), since biological or birth-assigned sex is different from gender and doesn't always tell your whole story or who you are. Gender is much more complex: it is a legal and social category, and a set of societal expectations about people's behaviour, characteristics and way of thinking.

In table 1S, the values of BMI of the women included normal BMI (20.8 kg/cm²) to women with obesity class 3 (44.0 kg/cm²), and by age included postmenopausal women,  that is important, because subcutaneous adipose tissue in women changes according to age and BMI.

Results section

The authors refer “MSCs were isolated from stromal-vascular fraction of subcutaneous adipose tissue of healthy 34 young donors that were taking no medications” and in the results mention that “The analysis of the spatial distribution of α1-adrenoceptor isoforms in adipose tissue of obese normotensive and hypertensive donors” What were the characteristics of the individuals with hypertension? And were taking medication for hypertension? If so, what kind of antihypertensives did they take? Or were of new diagnosis?

The authors analyzed the MSCs of women and men as one group? As well as, in the hypertensive group, did they mix subjects with normal BMI, overweight or obesity?

Discussion section:

Regarding to “This study provides novel evidence that MSCs within adipose tissue vasculature may be critically important in promoting life-threatening obesity complications, particularly obesity induced arterial hypertension” How the authors prove that hypertension was for obesity and not for other causes?

Author Response

Response to Reviewer 2 Comments

1) First, is better to use People-First Language, i.e. "individuals with obesity" rather than "obese individuals", generally, 'obese' should not be used as an adjective.

Response 1: We thank the esteemed Reviewer for this comment.  We have tried to use this vocabulary wherever the style of the text allows.

2) In table 1S, please change gender by sex (female or male), since biological or birth-assigned sex is different from gender and doesn't always tell your whole story or who you are. Gender is much more complex: it is a legal and social category, and a set of societal expectations about people's behaviour, characteristics and way of thinking.

Response 2: We have changed the term.

3) In table 1S, the values of BMI of the women included normal BMI (20.8 kg/cm²) to women with obesity class 3 (44.0 kg/cm²), and by age included postmenopausal women,  that is important, because subcutaneous adipose tissue in women changes according to age and BMI.

Response 3: According to our preliminary data, this parameter did not affect the ability of MSCs to alpha1A-adrenergic receptor sensitization after sympathetic overdrive modeling.

4) The authors refer “MSCs were isolated from stromal-vascular fraction of subcutaneous adipose tissue of healthy 34 young donors that were taking no medications” and in the results mention that “The analysis of the spatial distribution of α1-adrenoceptor isoforms in adipose tissue of obese normotensive and hypertensive donors” What were the characteristics of the individuals with hypertension? And were taking medication for hypertension? If so, what kind of antihypertensives did they take? Or were of new diagnosis?

Response 4: The diagnosis of obesity-associated hypertension was made based on the absence of causes of secondary hypertension and according to the patient's long history of obesity. Adipose tissue sampling was carried out in patients with obesity and a long-standing arterial hypertension. We have changed this incorrect phrase and added a supplementary file with description of antihypertensive therapy of the patients with obesity.

5) The authors analyzed the MSCs of women and men as one group? As well as, in the hypertensive group, did they mix subjects with normal BMI, overweight or obesity?

Response 5: In this study, we did not segregate patients by sex. According to our previous data, there are no significant differences between men and women in the molecular mechanisms of sensitization to noradrenaline. To analyze the correlation between arterial hypertension and the ability of MSCs to the sensitization of a1A-adrenoceptors, only material from patients with obesity (BMI > 30 kg/m2) was used. Patients with normal BMI or overweight were excluded. To elucidate the molecular mechanisms of a1A-adrenoceptors sensitization, patient MSCs capable of sensitizing adrenoceptors were selected. 

6) Regarding to “This study provides novel evidence that MSCs within adipose tissue vasculature may be critically important in promoting life-threatening obesity complications, particularly obesity induced arterial hypertension” How the authors prove that hypertension was for obesity and not for other causes?

Response 6: Donors of adipose tissue were recruited into the groups of hypertensive and normotensive patients with obesity according to the characteristics described in the Methods section. Causes of secondary hypertension were excluded according to the examination and questioning of patients. These patients also had a long history of obesity. According to literature data, 70% cases of primary arterial hypertension are associated with obesity (https://doi.org/10.1161/CIRCRESAHA.116.305697). And, on the other hand, weight loss often leads to normalization of blood pressure in hypertensive patients with obesity (https://doi.org/10.1097/HJH.0b013e3283537347). Thus, we can conclude that in the patients with obesity, the development of obesity is highly likely to be associated with obesity, and not with other causes.

Round 2

Reviewer 1 Report

The authors have taken into account our comments

Author Response

Thanks a lot for your review!

Reviewer 2 Report

The authors responded to most of my comments satisfactorily.

However, regarding to my observation “In table 1S, the values of BMI of the women included normal BMI (20.8 kg/cm²) to women with obesity class 3 (44.0 kg/cm²), and by age included postmenopausal women,  that is important, because subcutaneous adipose tissue in women changes according to age and BMI.

The authors answered: According to our preliminary data, this parameter did not affect the ability of MSCs to alpha1A-adrenergic receptor sensitization after sympathetic overdrive modeling”.

Have you published those results yet? o Do you plan to publish them in the future? If not, could the authors show us those results?

Regarding to my observation “The authors analyzed the MSCs of women and men as one group?

The authors answered: In this study, we did not segregate patients by sex. According to our previous data, there are no significant differences between men and women in the molecular mechanisms of sensitization to noradrenaline.

Have you published those results yet? o Do you plan to publish them in the future? If not, could the authors show us those results?
